# Quantitative Detection of Natural Rubber Content in *Eucommia ulmoides* by Portable Pyrolysis-Membrane Inlet Mass Spectrometry

**DOI:** 10.3390/molecules28083330

**Published:** 2023-04-10

**Authors:** Minmin Guo, Mingjian Zhang, Shunkai Gao, Lu Wang, Jichuan Zhang, Zejian Huang, Yiyang Dong

**Affiliations:** 1College of Life Science and Technology, Beijing University of Chemical Technology, Beijing 100029, China; 2Rubber Plant Research Center, Beijing University of Chemical Technology, Beijing 100029, China; 3College of Materials and Engineering, Beijing University of Chemical and Technology, Beijing 100029, China; 4Center for Advanced Measurement Science, National Institute of Metrology, Technology Innovation Center of Mass Spectrum for State Market Regulation, Beijing 100029, China

**Keywords:** *Eucommia ulmoides*, quantitative, pyrolysis-membrane inlet mass spectrometry, pyrolysis-gas chromatography, natural rubber

## Abstract

*Eucommia ulmoides* gum (EUG) is a natural polymer predominantly consisting of trans-1,4-polyisoprene. Due to its excellent crystallization efficiency and rubber-plastic duality, EUG finds applications in various fields, including medical equipment, national defense, and civil industry. Here, we devised a portable pyrolysis-membrane inlet mass spectrometry (PY-MIMS) approach to rapidly, accurately, and quantitatively identify rubber content in *Eucommia ulmoides* (EU). EUG is first introduced into the pyrolyzer and pyrolyzed into tiny molecules, which are then dissolved and diffusively transported via the polydimethylsiloxane (PDMS) membrane before entering the quadrupole mass spectrometer for quantitative analysis. The results indicate that the limit of detection (LOD) for EUG is 1.36 μg/mg, and the recovery rate ranges from 95.04% to 104.96%. Compared to the result of pyrolysis-gas chromatography (PY-GC), the average relative error is 1.153%, and the detection time was reduced to less than 5 min, demonstrating that the procedure was reliable, accurate, and efficient. The method has the potential to be employed to precisely identify the rubber content of natural rubber-producing plants such as *Eucommia ulmoides*, *Taraxacum kok-saghyz* (TKS), *Guayule*, and *Thorn lettuce*.

## 1. Introduction

Natural rubber (NR) is a significant industrial raw material and strategic material related to the national economy, people’s livelihoods, national defense and public security. Even though rubber can be synthesized using petrochemical products, natural rubber has unique properties that these synthetic materials cannot match [1,2]. *Hevea brasiliensis* is the only source of commercialized natural rubber. However, due to phenological constraints, the prevalence of South American Leaf Blight (SALB), and the occurrence of life-threatening “latex allergies”, people are forced to intensively explore alternative rubber-producing plants for natural rubber [3]. In response to the scarcity of natural rubber, several nations and organizations are actively carrying out research projects for alternative natural rubber to replace *Hevea brasiliensis* [4], such as the Rubber Dandelion Industry Technology Innovation Strategic Consortium and Eucommia Resources High-value Utilization Industry Technology Innovation Alliance of China, and the EU-PEARLS project in Europe [5,6,7,8].

*Eucommia ulmoides* Oliver (EU) is an elm-like deciduous tree belonging to the Eucommaceae and Eucommia genus [9] and is a unique and precious tree species in China. Moreover, EU is one of the tree species with the best combination of economic, social and ecological benefits. The EU genome contains numerous functional genes that confer resistance to cold, drought, salinity, pests, and diseases, making it ideal for improving the ecological environment of fragile areas and providing high ecological value in ecological and urban construction [10]. Furthermore, the bark, leaves and fruit of EU contain *Eucommia ulmoides* gum (EUG) and precious medicinal ingredients, which have high edible and medicinal value, and which have unique advantages in the treatment of hypertension, blood lipids, low back pain and metabolic diseases [11,12]. EUG is a natural gum extracted from the plant tissue of EU, and is a rare biopolymer resource in China [13]. EUG is abundant in the bark, root bark, and leaves of EU. Its primary component is trans-1,4-polyisoprene, which possesses a unique rubber-plastic duality [14]. It is utilized to develop high-performance, high-end rubber products such as submarine cables, medical devices, functional materials, and engineering tires. To realize the commercialization of EUG and obtain high-yielding rubber plants, quick and accurate detection of rubber content is a crucial step, and can promote the rapid development of natural rubber research.

Various analytical methods have been developed for the detection of natural rubber content, including differential gravimetric methods [15,16,17], accelerated solvent extraction (ASE) [18], refractive index measurements [19], Fourier transform infrared spectroscopy (FT-IR) [20,21], near infrared spectroscopy (NIRS) [22,23,24], gel permeation chromatography (GPC) [25], nuclear magnetic resonance (NMR) [26], and pyrolysis gas chromatography/mass spectrometry (PY-GC/MS) [27,28]. However, methods such as the differential gravity method and ASE consume a large amount of organic solvents. These procedures are cumbersome and time-consuming, the reproducibility is poor, and the detection throughput is low. The accuracy of the determination of the rubber hydrocarbon content by refractive index measurement is dependent on a number of factors, such as weighing and density measurement, which can easily cause large errors in the analysis results, and the accuracy and precision of the measurement are limited. Furthermore, infrared spectroscopy (IRS) and NIRS, as modeling approaches for predictive analysis, require a sizable quantity of referential sample data, and the difficulties of data collection will definitely lead to inconsistent measurement results. In addition, NMR and GPC require large-scale instruments, and the sample pretreatment is complicated and expensive, making them inappropriate for field testing. Furthermore, although PY-GC/MS provides more precise detection of rubber content, its operation and instrument maintenance costs are relatively high. All substances must enter the chromatographic column for separation, necessitating an additional analytical step and increasing the analysis time.

Membrane inlet mass spectrometry (MIMS) is a highly efficient separation technique that uses a semi-permeable membrane to isolate target substances from liquid or gas-phase systems, allowing for mass spectrometry analysis of complex samples without the need for chromatographic separation. MIMS can be divided into a membrane inlet, ion source, and mass analyzer based on its structural composition. The separation principle of MIMS is based on the different permeation rates of substances that are separated, due to differences in their molecular shape, size, and solubility in the membrane, when driven by the gas pressure difference across the permeable membrane. Substances with higher permeation rates are enriched on the side closer to the ion source, achieving the purpose of separation. In short, MIMS is a process of dissolving and diffusing the separated substances through the membrane. Most MIMS use a semi-permeable or selectively permeable membrane as the separation method, and select small, portable quadrupole or ion trap mass spectrometers for mass analysis.

MIMS was developed by Hoch and Kok in 1963; the work demonstrated the application of this system to the study of the kinetics of oxygen release in photosynthesis [29]. The purpose of the invention is to selectively monitor the concentration of gases and volatile organic compounds (VOCs) in biological systems without disrupting its function or requiring sample handling. The membrane inlet can directly interface with virtually any chemical or biological process without interference, while introducing biologically unimportant gases and low molecular VOCs into the mass spectrometer for selective detection [30]. Nowadays, MIMS technology has been widely applied in the detection of pollutants in environmental samples [31], ecology [32], and microbial metabolism engineering [33]. MIMS has evolved from an analytical method for atmospheric gases dissolved in water to a mature on-site portable technology used to analyze VOCs in solid, liquid, and gas samples without prior sample preparation.

A membrane inlet system was created by Vreeken and Houriet in 1995 to enable the coupling of atmospheric pyrolysis with a mass spectrometer [34]. In their research, a new membrane interface was developed and used to couple pyrolysis and mass spectrometry based on the study of MIMS. Silicon membranes were used. Utilizing the interface, a pyrolysis unit and an ion-trap mass spectrometer were coupled directly.

Alan [35] also successfully used PY-MIMS technology to study the oxidation products of several peptides in an air-buffered quadrupole ion trap mass spectrometer. The results show that when complex mixtures are analyzed using the membrane entrance of the ion trap mass spectrometer system, identification becomes much more challenging because the ion molecules in the ion trap react to create peculiar peaks that are not observed with other kinds of mass spectrometers.

In this paper, we present a new quantitative method of PY-MIMS, for rapidly determining the rubber content of leaves and bark in EU. This method uses a quadrupole as the mass analyzer of MIMS, and the membrane material is PDMS membrane, which has an excellent separation effect on organic matter. In this method, the polyisoprene component in EU is pyrolyzed into tiny molecular substances, which are directly analyzed by MIMS for the determination of the rubber content in leaves and bark of EU and comparison using PY-GC. The results showed that PY-MIMS could be utilized for the rapid and high-throughput quantitative determination of rubber content in EU.

## 2. Results

### 2.1. Selection of PY-MIMS Calibration Curve Type

Pyrolysis-mass spectrometry (PY-MS) technology has been used in the qualitative analysis and detection of polymers such as microplastics [36]. Generally, PY-GC/MS is used for qualitative analysis in the petrochemical industry, food safety, medicine, public security, and other fields [37,38,39]. Up to now, only a little research has utilized mass spectrometry directly in the quantitative analysis of polymers. A recent study [40] showed that a custom-built electromagnetic heating pyrolyzer coupled to mass spectrometry can be used for rapid quantitative analysis of nanoplastics. Pyrolysis, as an online sample pretreatment method for organic macromolecules, is directly combined with MIMS technology for quantitative analysis of polymers, which is a very challenging task in the field of mass spectrometry.

PY-MIMS on ten EUG reference material (RM) injection samples of different masses was performed herein, and the peak area and peak intensity corresponding to *m/z* 68 ion in the mass spectrum were calculated. As shown in Appendix A, the mass (mg) of the EUG RM is used as the abscissa to draw a calibration curve. Appendix A corresponds to the peak area calibration curve A: y = 1.0063x + 0.0187, and the correlation coefficient is 0.9986; Appendix A corresponds to the peak intensity calibration curve B: y = 4.4483x + 0.4468, and the correlation coefficient is 0.9994. Observably, the linear relationship between the two curves is satisfactory and conforms to the quantitative criteria. However, the correlation coefficient of curve B is better, and the data processing method using peak intensity for on-site mass spectrometers can intuitively achieve real-time detection as well. Therefore, during the actual measurement, the peak intensity parameters were selected to draw a calibration curve for subsequent detection of the rubber content of EU.

### 2.2. PY-GC Calibration Curve Establishment

As a mature quantitative analysis method, PY-GC has been used in the content determination of natural rubber. In this paper PY-GC is used as the reference method for PY-MIMS. Due to the column’s resolution, it is impossible to fully separate the isoprene monomer from substances such as water molecules in the air, and the response of isoprene is often relatively low. Therefore, the actual quantitative process selects the peak area of isoprene dimer limonene to quantify EUG.

Ten EUG RM samples with different qualities were injected into PY-GC. After being pyrolyzed into small molecules by a pyrolyzer, they enter the chromatographic column under the push of nitrogen to achieve separation, and the responsive signal is collected and recorded by a thermal conductivity detector to obtain a chromatogram. The peak area of the characteristic product limonene with retention time of 9 min in the chromatogram was calculated, taking it as the ordinate and the mass (mg) of EUG RM as the abscissa, and drawing the calibration curve. As shown in Appendix A, the calibration curve acquired for chromatographic quantification is y = 2513.5x − 65.042, and the correlation coefficient is 0.9990. This curve also reaches the criteria required for quantification and can be used for the quantitative detection of the rubber content of EUG.

### 2.3. Determination of Rubber Content in EU Samples

EU is a dioecious plant and EUG mainly exists in the leaves, bark, and seeds of EU. No theoretical research has yet shown the difference in rubber content between male and female trees of EU. To reduce the variability among samples, the samples used in this study were all from Eucommia male trees, instead of using mixed samples. The mass spectrum of EU leaves measured by PY-MIMS is shown in Figure 1.

Quantitative detection of 10 samples each from EU leaves and bark was carried out by PY-GC and PY-MIMS respectively, and then the results were compared and analyzed. Among them, YA, EA, HA, SA, WA represent different samples of EU leaves, and YB, EB, HB, SB, WB represent different samples of EU bark. Each sample was measured three times. The measurement results are tabulated in Table 1, showing that the range of rubber content detected by PY-GC is 1.510% ± 0.080%~8.230% ± 0.209%, while the range of rubber content detected by PY-MIMS is 1.462% ± 0.152%~8.021% ± 0.392%; the average relative error of the two methods is only 1.153%. Compared with the gravimetric method, the detection error is greatly reduced, which proves that the determination result of the PY-MIMS method is reliable. At the same time, the direct combination of thermal cracker and mass spectrometry eliminates the chromatographic column separation step in the quantitative analysis of polymers, thereby greatly shortening the detection time, and enabling faster response time, more speed and better efficiency.

### 2.4. Determination of Limit of Detection and Recovery Rate

The limit of detection (LOD) is a comprehensive indicator for evaluating the sensitivity of a method, and it is also the main technical indicator for evaluating instrument performance and analytical methods. To calculate the LOD of this method, the response value of the blank samples was measured fifteen times. The *m/z* 68 ion was extracted and the peak intensity was calculated to obtain the standard deviation of the peak intensity of fifteen blank samples. Then, three times the standard deviation was divided by the slope of the calibration curve to obtain the LOD of the method, which was 1.36 μg/mg. Compared with PY-GC (2.603 mg/g) [28], the LOD was lower.

The accuracy of the method was evaluated by recovery tests which were carried out by spiking samples with different EUG RM masses at three content levels. After accurately measuring 10 μL, 30 μL, and 50 μL of 5 μg/μL EUG RM stock solution (i.e., the spiked quantity is 0.05 mg, 0.15 mg, and 0.25 mg of EUG RM), PY-MIMS detection was performed. The peak intensity of the *m/z* 68 ion of the rubber quantitative characteristic ion was extracted and calculated, substituting the corresponding peak intensity into the quantitative calibration curve y = 4.4483x + 0.4468, obtaining the rubber content in the EU sample, and finally obtaining the recovery rate of the sample. Each group was measured five times in parallel. The recovery results are shown in Table 2. The recoveries of the samples were all between 95.04% and 104.9%, and the mean RSD was 5.96%.

## 3. Materials and Methods

### 3.1. Experimental Reagents and Plant Materials

Toluene, acetone, and methanol of analytical grade were purchased from Sinopharm Holding Chemical Reagent Co. Ltd. in Beijing, China. 

EU materials were taken from *Eucommia ulmoides* Park in Beijing. The collected Eucommia leaf, fruit, and bark samples were dried in a vacuum drying oven at 60 °C for 12 h and dried to constant weight. The dried Eucommia samples were pulverized with a pulverizer until they were utterly powdered, then placed in airtight plastic bags, and stored in a desiccator at room temperature until used.

### 3.2. Experimental Equipment and Conditions

The PY-MIMS used in the experiment consists of a PY3030S singleshot thermal cracker (Frontier, Tokyo, Japan), a membrane inlet system, and an online quadrupole mass spectrometer (National Institute of Metrology, Beijing, China). The membrane material is PDMS, and high-purity nitrogen is used as the carrier gas. The structural schematic diagram is shown in Figure 2.

Pyrolysis conditions: thermal pyrolysis temperature 550 °C, injection port temperature 300 °C, pressure stabilization time 20 s, cleaning time 10 s, and pyrolysis time 0.1 s.

Mass spectrometry conditions: inlet membrane temperature 90 °C, ion source temperature 120 °C, filament electron energy −65 eV, filament current feedback 2.2 A, repeller voltage −5 V, ion lens 1 voltage −5 V, ion lens 2 voltage −15 V, ion lens 3 voltage −180 V; multiplier voltage −1150 V, high-end compensation voltage 1.005 V, low-end compensation voltage 0.15 V, quadrupole float voltage −6.00 V; carrier gas flow rate 150 mL/min, pumping speed of the split air pump 190 mL/min. The vacuum environment is kept at 10^−6^ Pa, and the entire sampling process takes 3–4 min.

### 3.3. PY-MIMS Detection Principle

The pyrolysis of EUG generates a substantial quantity of isoprene monomer and limonene dimer, which produce a high intensity at *m/z* 68 ion in mass spectrometry, so this is selected as the quantitative characteristic ion of EUG (Figure 3). Under the same conditions of PY-MIMS, there is a linear relationship between the rubber mass of EUG and the peak intensity and peak area of quantitative characteristic product ions.

First, EUG calibrators of EU reference material with different mass ratios were prepared for injection, the peak area and peak intensity of the quantitative characteristic ion generated in the mass spectrometer for each sample were calculated, and the quantitative calibration curve of EUG was drawn with the sample mass as the abscissa. Second, various EU samples were tested by PY-MIMS. Ultimately, by substituting the peak area or peak intensity data into the calibration equation, the calculation of the rubber content in EU samples can be realized.

### 3.4. PY-MIMS Detection Steps

#### 3.4.1. Preparation of EUG Reference Material by Soxhlet Extraction

1 g of the aforementioned EU fruit powder was weighed into a 100 mL cellulose thimble, and an appropriate amount of diatomaceous earth was added into the cellulose thimble. The cellulose thimble containing the mixed sample powder was put into a 150 mL Soxhlet extractor. The highly polar solvent acetone was utilized to extract polar compounds such as resin in the heating mantle for about 6–8 h. The cellulose thimble was taken out and dried to a constant weight in a vacuum drying oven at 70 °C and then the low polarity solvent toluene was used to extract EUG for about 6–8 h in a heating mantle.

EUG was precipitated by concentrating the extract in a rotary evaporator to a tiny amount, then 4–5 times the volume of methanol was added. After removing the supernatant, the precipitated EUG was dissolved, and the precipitation process was repeated until the extract became translucent and transparent. The final precipitate of snow-white EUG was dried to a constant weight in a vacuum oven at 45 °C to obtain a EUG RM.

#### 3.4.2. Plotting the PY-MIMS Calibration Curve

After placing 50 mg of the above-mentioned EUG RM in 7 mL of toluene, the mixture was sonicated for 4 h until the EUG RM was completely dissolved, then diluted in a 10 mL volumetric flask to get a stock EUG RM concentration of 5 μg/μL. A pipette was used to accurately transfer 2, 4, 10, 20, 30, 40, 50, 60, 70, and 80 μL of EUG RM solution, respectively, into the pyrolyzer sample cups. The weighed sample cups with EUG RM were dried in a vacuum oven at 45 °C for 30 min, and the injection masses used as EUG calibrators were 10, 20, 50, 100, 150, 200, 250, 300, 350, and 400 μg, respectively.

After the toluene was dried, the sample cups were placed in the pyrolysis injection port in turn for PY-MIMS detection and to acquire the corresponding mass spectrum. The *m/z* 68 ion was extracted, and the peak area and peak intensity were calculated, respectively. The mass (mg) of the EUG RM was taken as the abscissa to draw the quantitative calibration curve.

Py-MIMS were then performed for the EU samples. After pyrolysis, the peak intensity or peak area of the pyrolysis product *m/z* 68 ion under the experimental conditions was substituted into the calibration curve to calculate the rubber content in the sample.

### 3.5. PY-GC Detection

PY-GC detection is the quantitative determination of rubber content based on the linear relationship between the peak area of the pyrolysis product and the quantity of EUG RM [28]. According to the external calibration method, different amounts of EUG RM were weighed, a calibration curve was drawn, and various EU samples were tested by Py-GC. The rubber content in the EU samples can be easily calculated by substituting the peak area data into the calibration equation.

The experiments used an Agilent 6890 N GC equipped with a PY UA-5 capillary column (30.0 m × 250 μm × 0.25 μm) and a thermal conductivity detector (TCD). The pyrolysis conditions included a pyrolysis temperature of 550 °C and a pyrolysis time of 0.1 s. Gas chromatography conditions included an inlet temperature of 250 °C and a split ratio of 50:1. The programmed column temperature ramp was increased from 40 °C (after a 5 min hold) to 130 °C at a rate of 10 °C/min, followed by a 10 min hold, and then increased to 280 °C at a rate of 10 °C/min with a 10 min final hold [28].

## 4. Conclusions

In this study, a lab-developed portable pyrolysis mass spectrometer was developed, and the PY-MIMS technology was proposed for the first time for the determination of EU rubber content. This technique enables high-throughput, fast, and real-time screening of natural rubber. The LOD for EUG was 1.36 μg/mg, and the recovery rates were between 95.04% and 104.93%, which proved that the sensitivity and accuracy of the method were satisfactory. In terms of judging the compatibility of this method, we compared it with the established PY-GC method. The average relative error of rubber content in the tested samples was 1.153%, and a reliable result with good consistency was thus obtained.

In summary, an accurate and efficient quantitative detection technology for natural rubber content characterization has been successfully developed to assist EU germplasm screening practices and help realize the large-scale industrial production of EU, in order to acquire EUG as an alternative natural rubber. Compared with the existing detection methods, the new technology has higher sensitivity and precision, and at the same time significantly reduces the test time and difficulty of operation. The establishment of this new technology solves the problem of being unable to detect the rubber content of EU in real time, quickly and conveniently in the actual production and planting processes. It meets the need for accurate real-time quantitative detection technology in the field of rubber detection. The method can be applied as well to the rapid and high-throughput quantitative detection of rubber content in other natural rubber-containing plants, such as TKS, *Guayule*, and *Thorn lettuce* for a sustainable development of the rubber industry.

## Figures and Tables

**Figure 1 molecules-28-03330-f001:**
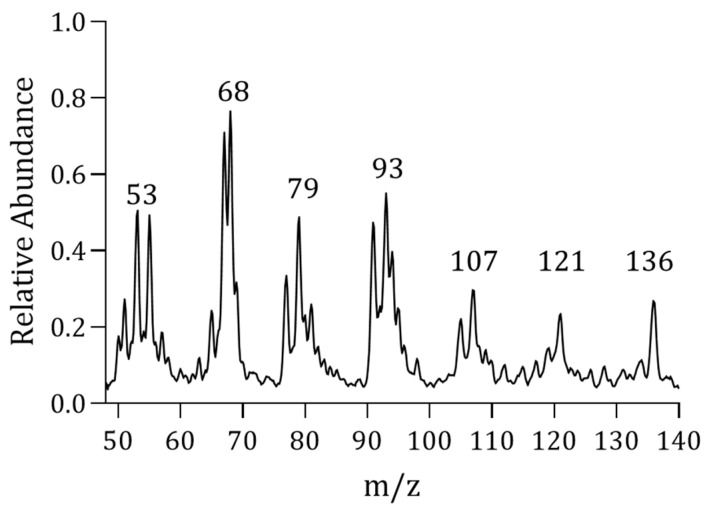
PY-MIMS spectrum of EU leaves.

**Figure 2 molecules-28-03330-f002:**
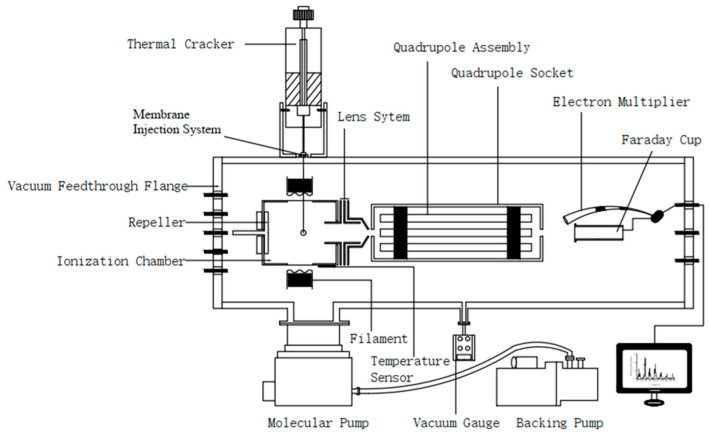
Schematic diagram of the structure of the PY-MIMS.

**Figure 3 molecules-28-03330-f003:**
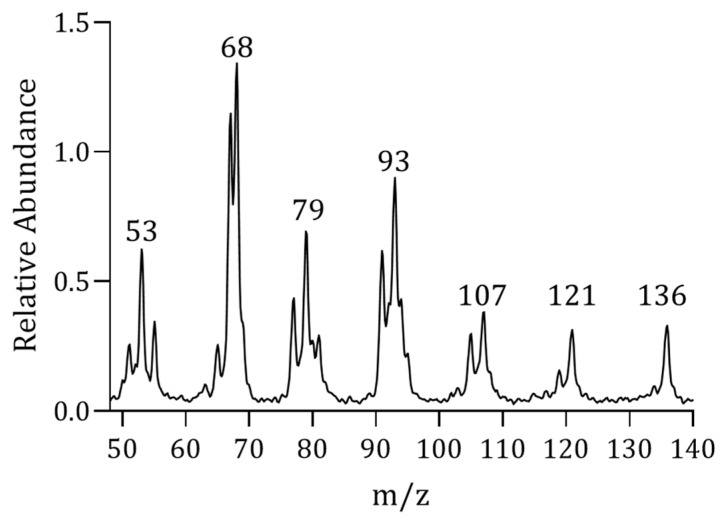
PY-MIMS spectrum of EUG.

**Table 1 molecules-28-03330-t001:** Comparison of Rubber Content in Samples by PY-GC and PY-MIMS.

EU Variety	PY-GC Determined Rubber Content Mass Fraction ± SD, %	Confidence Interval, 95%	PY-MIMS Determined Rubber Content Mass Fraction ± SD, %	Confidence Interval, 95%	Relative Error *, %
YA	1.510 ± 0.080	1.510 ± 0.198	1.462 ± 0.152	1.462 ± 0.464	−3.178
EA	2.560 ± 0.116	2.560 ± 0.289	2.425 ± 0.201	2.425 ± 0.611	−5.308
HA	2.089 ± 0.118	2.089 ± 0.293	2.121 ± 0.109	2.121 ± 0.333	1.526
SA	2.867 ± 0.387	2.867 ± 0.961	2.915 ± 0.141	2.915 ± 0.430	1.660
WA	2.613 ± 0.160	2.613 ± 0.398	2.755 ± 0.277	2.755 ± 0.844	5.416
YB	6.603 ± 0.564	6.603 ± 1.401	6.358 ± 0.283	6.358 ± 0.861	−3.714
EB	8.230 ± 0.209	8.230 ± 0.519	8.021 ± 0.392	8.021 ± 1.193	−2.533
HB	4.905 ± 0.292	4.905 ± 0.725	5.016 ± 0.285	5.016 ± 0.867	2.262
SB	4.679 ± 0.265	4.679 ± 0.657	4.508 ± 0.227	4.508 ± 0.692	−3.659
WB	4.752 ± 0.206	4.752 ± 0.512	4.561 ± 0.250	4.561 ± 0.761	−4.001

Relative error * = (PY-MIMS rubber Mean Content—PY-GC rubber Mean Content)/PY-GC rubber Mean Content × 100.

**Table 2 molecules-28-03330-t002:** Recovery results of EUG spiking experiment.

EUG Spiked Amount, mg	Quantity Recovered, mg	Confidence Interval, 95%	Recovery, %	RSD, %
0.05	0.049 ± 0.005	0.049 ± 0.007	97.84	9.84
0.15	0.157 ± 0.005	0.157 ± 0.007	104.93	3.33
0.25	0.238 ± 0.011	0.238 ± 0.016	95.04	4.72

## Data Availability

The data presented in this study are available on request from the corresponding author.

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
