# Peer review of "Quantitative Detection of Natural Rubber Content in Eucommia ulmoides by Portable Pyrolysis-Membrane Inlet Mass Spectrometry"

_molecules, 2023, doi:10.3390/molecules28083330_

Round 1

Reviewer 1 Report

This is a very interesting paper using pyrolysis membrane introduction mass spectrometry for quantitative identification of the rubber content in Eucommia ulmoides. The described application is novel and of general interest.

It is not the first time MIMS is used as a detector for pyrolysis and papers on the topic have been published decades ago. The manuscript should have a reference to earlier PY-MIMS work and I can recommend the original by Vreeken and Houriet, Analytica Chimica Acta 313 (1995) 237-241.

Further the word "injection" in the abbreviation of MIMS should be replaced by "introduction" or "inlet". The membrane interface described here is basically the same as those used by the MIMS community and it will make a lot of confusion to introduce a third name for the same technique.

The manuscript has a wording problem page 4 line 129-131 "Quantitative detection of 10 samples from two parts of Eucommia ulmoides leaves and bark was carried out by PY-GC and PY-MIMS respectively, and then the results were compared and analyzed (Fig. 3)". The Figure just show a single mass spectrum of EU leaves and nothing that compares PY-GC and PY-MIMS.

Reviewer 2 Report

Rubber content in Eucommia ulmoides is determined by a pyrolysis-membrane injection mass spectrometry. Pyrolyzing and diffusing produce charged molecules. Then these molecules are analyzed by a quadrupole MS. The pyrolysis ionization and quadrupole techniques are commonly used in MS field. The detected object rubber is neither a complex nor interesting material.

Reviewer 3 Report

On page 3, line 86, quantitative statement should be added after qualitative statement.

·         it should be stated in the text that the PY-GC/MS method was chosen as the reference method and the relative error was calculated accordingly.

·         Relative error values in Table 1 should be checked.

·         The LOD value of the PY-GC/MS method should be stated in the text. (Page 5, Line 155)

·         In Table 1 and Table 2, the results can be given with a certain level of confidence.

·         In the keywords section, the full name may be written instead of abbreviations. For example: PY-MIMS, PY-GC,

·         There is no space between some words in the 2nd page, the 2nd paragraph, it should be added.

·         Since the abbreviation of the expression "pyrolysis-gas chromatography/mass spectrometry" on page 86, 3rd page has been given in the previous sections, its abbreviation should be used.

·         What does the abbreviation RM on page 3, line 90 refer to?

Round 2

Reviewer 2 Report

I still hold the first review opinion.

Author Response

It is necessary to explain again that domestic analytical instruments are heavily dependent on imports and face the risk of "stuck" in key components. In this study, a self-developed domestic portable pyrolysis-membrane sampling mass spectrometer specially developed for rubber detection is used to contribute to China's development of related instruments.

At the same time, PY-MIMS with the hyphenation of pyrolyzer and membrane intel mass spectrometer is a relatively cutting-edge quantitative analysis technology in the current research. This study successfully developed an accurate and efficient quantitative detection technology for the characterization of natural rubber content and expects to apply the technology to the quantitative detection of other natural rubber biopolymers to provide technical support for the sustainable and healthy development of my country's rubber industry.

We appreciate the reviewer's comments again.